Intrinsic factors behind long COVID: exploring the role of nucleocapsid protein in thrombosis

Eltayeb Ahmed 1
Adilović Muhamed 2
Golzardi Maryam 2
Hromić-Jahjefendić Altijana 2
Rubio-Casillas Alberto 3
Uversky Vladimir N. vuversky@usf.edu 4
Redwan Elrashdy M. lradwan@kau.edu.sa 1 5
1 Department of Biological Sciences, Faculty of Science, King Abdulaziz University , Jeddah , Saudi Arabia
2 Department of Genetics and Bioengineering, Faculty of Engineering and Natural Sciences, International University of Sarajevo , Sarajevo , Bosnia and Herzegovina
3 Autlan Regional Hospital, Jalisco Health Services , Jalisco , Mexico
4 Department of Molecular Medicine and USF Health Byrd Alzheimer’s Research Institute, Morsani College of Medicine, University of South Florida , Tampa , FL , United States of America
5 Therapeutic and Protective Proteins Laboratory, Protein Research Department, Genetic Engineering and Biotechnology Research Institute, City of Scientific Research and Technological Applications , New Borg EL-Arab , Alexandria , Egypt
Nakai Kenta
Electronic publication date: 2025 May 20
Publication date: 2025
Volume: 13
Electronic Location ID: e19429
Received 2024 Aug 9; Accepted 2025 Apr 15
Copyright: ©2025 Eltayeb et al.
Copyright year: 2025
Copyright holder: Eltayeb et al.
License: This is an open access article distributed under the terms of the Creative Commons Attribution License, which permits unrestricted use, distribution, reproduction and adaptation in any medium and for any purpose provided that it is properly attributed. For attribution, the original author(s), title, publication source (PeerJ) and either DOI or URL of the article must be cited.
License URL: https://creativecommons.org/licenses/by/4.0/

Keywords: SARS-CoV-2, Endothelial cells, Nucleocapsid protein, Long COVID, Thrombosis

Funding: The authors received no funding for this work.

==============================
COVID-19, caused by the SARS-CoV-2, poses significant global health challenges. A key player in its pathogenesis is the nucleocapsid protein (NP), which is crucial for viral replication and assembly. While NPs from other coronaviruses, such as SARS-CoV and MERS-CoV, are known to increase inflammation and cause acute lung injury, the specific effects of the SARS-CoV-2 NP on host cells remain largely unexplored. Recent findings suggest that the NP acts as a pathogen-associated molecular pattern (PAMP) that binds to Toll-like receptor 2 (TLR2), activating NF-κB (nuclear factor kappa-light-chain-enhancer of activated B cells) and MAPK (mitogen-activated protein kinase) signaling pathways. This activation is particularly pronounced in severe COVID-19 cases, leading to elevated levels of soluble ICAM-1 (intercellular adhesion molecule 1) and VCAM-1 (vascular cell adhesion molecule 1), which contribute to endothelial dysfunction and multiorgan damage. Furthermore, the NP is implicated in hyperinflammation and thrombosis—key factors in COVID-19 severity and long COVID. Its potential to bind with MASP-2 (mannan-binding lectin serine protease 2) may also be linked to persistent symptoms in long COVID patients. Understanding these mechanisms, particularly the role of the NP in thrombosis, is essential for developing targeted therapies to manage both acute and chronic effects of COVID-19 effectively. This comprehensive review aims to elucidate the multifaceted roles of the NP, highlighting its contributions to viral pathogenesis, immune evasion, and the exacerbation of thrombotic events, thereby providing insights into potential therapeutic targets for mitigating the severe and long-term impacts of COVID-19.

Introduction

A deadly coronavirus known as SARS-CoV-2 (severe acute respiratory syndrome-coronavirus-2) emerged for the first time in December 2019. It resulted in the coronavirus disease 2019 pandemic (COVID-19), an acute respiratory infection that posed a worldwide health threat (Hu et al., 2021). From its origin, SARS-CoV-2 has been a subject of a giant array of studies covering a wide spectrum of topic areas, including research on the signalosome (Lundstrom et al., 2023a), treatment strategies (Lundstrom et al., 2023b), and its possible relevance to other illnesses (Hromić-Jahjefendić et al., 2022; Hromić-Jahjefendić et al., 2023; Golzardi et al., 2024). The virus can spread through multiple routes, including oral transmission, but the main mode of transmission is through expired droplets and aerosols from humans (Berentschot et al., 2024; Ballouz et al., 2023; Franco et al., 2024; Brodin, 2021). The SARS-CoV-2, when infects humans, causes a disease commonly referred to as “COVID-19,” which can manifest as anything from no symptoms to pneumonia, with the most severe outcome of COVID-19 being respiratory failure and/or cytokine storm resulting in death. The initial pathological changes induced by the virus are caused by its ability to bind to the ACE-2 (angiotensin-converting enzyme 2) receptor abundantly expressed on endothelial cells, which allows it to enter these cells and replicate. Approximately 60% of individuals infected with the virus seem to have cleared it from their system after 28 days and are able to return to their usual activities. Nonetheless, over 40% of people have several kinds of post-COVID complications, such as the one termed as “long COVID-19”.

Thrombosis has emerged as a significant complication of COVID-19, characterized by an increased incidence of both venous thromboembolism (VTE) and arterial thromboses. The pathophysiology of COVID-19-associated thrombosis involves complex interactions between the immune response, endothelial dysfunction, and the coagulation cascade, leading to a prothrombotic state that can result in severe clinical outcomes (Mizurini et al., 2021). Patients with COVID-19 often exhibit elevated levels of D-dimer (a dimer of small fibrin protein fragments present in the blood after a blood clot is degraded by fibrinolysis) and other biomarkers indicative of coagulopathy, which correlate with disease severity (Gorog et al., 2022). Despite the extensive research on thrombosis in COVID-19, gaps remain in understanding the long-term implications of these thrombotic events, particularly in relation to long COVID. Many individuals experience persistent symptoms following recovery, suggesting that the mechanisms underlying thrombotic complications may contribute to these ongoing health issues (Knight et al., 2022).

At the molecular level, when a cell is infected with SARS-CoV-2, the virus hijacks the cell’s mitochondrial function. Impairment of normal mitochondrial function results in the cytosolic accumulation of mitochondrial DNA, reflecting breakdown of the mitochondrial integrity. This type of DNA triggers the activation of inflammasomes and suppresses both innate and adaptive immunity within the body. Immunothrombosis is believed to play a significant role in COVID-19-associated coagulopathy (CAC) and the thrombogenic nature of COVID-19. This process is influenced by endothelial dysfunction, innate immune cell activation, NETosis (a cell death pathway involving formation of neutrophil extracellular traps (NETs)), platelet and complement activation, and the coagulation system.

A main player in COVID-19 pathogenesis is the nucleocapsid protein, which is crucial for viral replication and assembly. The nucleocapsid protein (NP) and viral RNA enter the host cell after infection to aid in replication and initiate the process of virus particle assembly and release (Narayanan et al., 2003). The virus infects cells expressing the ACE-2 enzyme on their plasma membrane, hijacks the cells’ synthetic macromolecules, and rapidly replicates and sheds. An individual infected with the virus may not show symptoms for the first 2–3 days, despite high levels of replication and shedding. Such individuals are termed “asymptomatic spreaders” (Brodin, 2021). Following that the host’s immune cells attack infected cells, releasing autacoids that cause symptoms such as cough, fever, and headache.

This review paper aims to explore the role of the nucleocapsid protein in platelet function related to COVID-19 and its post-acute sequelae, while also highlighting potential areas for future research. It is intended for researchers, clinicians, and healthcare professionals focused on the investigation and management of COVID-19 and its lingering complications.

Survey Methodology

We performed a literature review utilizing the PubMed database. Our search method included important phrases such as “SARS-CoV-2,” “COVID-19,” “Nucleocapsid protein,” “Platelet function,” “Immunothrombosis,” “Long COVID,” and “Coagulopathy” utilizing Boolean operators. We considered in total 157 peer-reviewed original research papers, reviews, and clinical trials published between December 2019 and the present (2024), limiting our selection to English-language literature. Articles were chosen based on their relevance to the role of the nucleocapsid protein in platelet function and its implications for COVID-19 development and post-acute complications. Non-peer-reviewed publications, editorials, opinion pieces, and conference abstracts were eliminated to ensure rigor. Data extraction entailed carefully assembling study objectives, techniques, findings, and conclusions, with a focus on quality and relevance. This strategy aims to give a full, up-to-date understanding of NP function in COVID-19.

The Nucleocapsid Protein

The genetic code of SARS-CoV-2 is composed of around 30,000 nucleotides that specify the production of four key structural proteins: the S, E, M, and NP (Lu et al., 2020). Of the four structural proteins, the NP is particularly immunogenic and is expressed in large quantities during infection. According to serological studies, NP antibodies in the blood plasma of COVID-19 patients are more sensitive and durable than antibodies against other structural proteins from SARS-CoV (Shi et al., 2003; Tan et al., 2004). Additionally, it has been documented that anti-N antibodies can be detected with great specificity in the early stages of infection (Leung et al., 2004; Li & Li, 2021). After infection, the viral RNA penetrates the host cell accompanied by the NP, which facilitates genome encasing and assembly, viral particle replication, and the emergence of newly generated virions (Narayanan et al., 2003; Zinzula et al., 2021).

There are three highly conserved regions in the SARS-CoV-2 NP. An intrinsically disordered region (IDR, i.e., region that does not have a unique structure but plays a number crucial functional roles) (residues 175-246) that contains an SR-rich domain (SRD, residues 176-206) connects the two of the ordered domains, the N-terminal domain (NTD, residues 48-174) responsible for RNA binding and the C-terminal domain (CTD, residues 247-364), which mediates oligomerization and also binds RNA (Huang et al., 2004; Hurst, Koetzner & Masters, 2009; Cong et al., 2020). SRD is particularly important, since it is the location of post-translational modifications, specifically phosphorylation (Fung & Liu, 2018). Moreover, arginine residues R95 and R177 are methylated, which is an important step in the viral packaging of SARS-CoV-2 (Cai et al., 2021). Figure 1 shows some of the structural and functional features of the NP and illustrates that this protein contains more than an eye can see. Here, Fig. 1A represents the per-residue intrinsic disorder profile of the SARS-CoV-2 NP and shows that this protein contains high levels of predicted intrinsic disorder (i.e., regions located above the disorder threshold of 0.5), possessing three highly disordered regions, the N-tail (residues 1-100, note that the C-terminal half of this disordered region covers 40% of the NTD as well), linker domain (residues 175-246), and C-tail (residues 365-419). In other words, ∼55% of this protein is predicted to be disordered. Importantly, these predictions are supported by the 3D structural model generated by AlphaFold, as Fig. 1B clearly shows that a very significant part of this protein is modeled with low confidence (red segments) and does not contain elements of ordered secondary structure. However, oligomerization of the SARS-CoV-2 NP induces formation of a stable structure in its CTD. This is illustrated by Fig. 1C representing the resulting structure that shows a compact, strongly interwoven dimer with a core four-stranded β-sheet that forms the majority of the dimer interface. The intrinsically disordered nature of the NP defines its multifunctionality and binding promiscuity. In fact, this protein interacts with various viral and host proteins to facilitate the SARS-CoV infection (see Fig. 1D). Genome packaging is done together with the matrix protein (M protein). It also interacts with nonstructural protein 3 (NSP3) which allows it to participate in replication-transcription complexes where RNA replication and transcription happen (Cong et al., 2020; Ye et al., 2020; Zinzula et al., 2021). Additionally, even though NP is mainly found in the cytoplasm (McBride, Van Zyl & Fielding, 2014), a portion of its copies can be localized in the nucleolus, which, together with ADP-ribosylation (Grunewald et al., 2018; Fung & Liu, 2018), could allow it to have an effect on the cell cycle (Su et al., 2020).

Figure 1 Structural and functional characterization of SARS-CoV-2 N protein.

(A) Per-residue intrinsic disorder profile generated for the SARS-CoV-2 N protein (UniProt ID: P0DTC9) generated by the RIDAO platform that assembles the outputs of several commonly used per-residue intrinsic disorder predictors in one plot (Dayhoff & Uversky, 2022). In addition to showing the outputs of PONDR® VLXT, PONDR® VL3, PONDR® VSL2, PONDR® FIT, IUPred_long and IUPred_short, RIDAO represents the mean disorder profile (MDP), which is calculated by averaging the disorder profiles of individual predictors. The light pink shade represents the MDP error distribution. The thin black line at the disorder score of 0.5 is the threshold between order and disorder, where residues/regions above 0.5 are disordered, and residues/regions below 0.5 are ordered. The dashed line at the disorder score of 0.15 is the threshold between order and flexibility, where residues/regions above 0.15 are flexible, and residues/regions below 0.15 are highly ordered. (B) 3D structure of the full-length protein modeled by AlphaFold (Jumper et al., 2023). Regions with low per-residue confidence scores (red) are expected to be disordered in isolation, whereas the two known domains are predicted with a high confidence (blue). (C) 3D structure of the dimeric form of the CTD of SARS-CoV-2 nucleocapsid protein (residues 247-364) (PDB: 6WZO) generated using PyMol (Schrödinger, 2015; Ye et al., 2020). (D) Protein-protein interaction network centered at the SARS-CoV-2 N protein. This network was generated by IntAct, which is an open-source, open data molecular interaction database populated by data either curated from the literature or from direct data depositions (Orchard et al., 2014). The network represents both viral and host proteins with established physical associations.

The Role of Nucleocapsid Protein in SARS-CoV-2 Life Cycle

Viral core formation

SARS-CoV-2 viral RNA synthesis and packaging are encoded by its genomic RNA (gRNA) that has a length around 30 kb long. This gRNA encodes the structural proteins mentioned before. NP, as a very commonly found structural protein in infected cells by COVID-19 is located in the core component of the virus (Bai et al., 2021). This protein is essential for virus replication, evading host immunity, and virus maturation (McBride, Van Zyl & Fielding, 2014; Peng et al., 2020; Stuwe et al., 2024). Additionally, one of the primary roles of the NP is to bind, condense, and package the viral genome. It does this by wrapping the viral RNA into structures called nucleocapsids or capsids, which are flexible, elongated, helical complexes made of ribonucleoprotein (RNP) (Zhao et al., 2021b; Wu et al., 2023; Stuwe et al., 2024). The nucleocapsid shields the genome and guarantees its punctual replication and consistent dissemination. The elongated nucleocapsids, with diameters ranging from 10 to 15 nm and lengths of several hundred nanometers, can be seen with an electron microscope (De Haan & Rottier, 2005). NP-RNA interactions and intermolecular relationships between disulfide-linked NP multimers occur within the nucleocapsid (Robbins et al., 1986). The interaction between the NP and RNA is facilitated by binding signals present in the leader RNA sequences (Baric et al., 1988). In the course of the virus’s life cycle, several NP molecules interact with both gRNA and subgenomic RNA (sgRNA), implying a function for the NP in the transcription and translation of the virus (Baric et al., 1988; Narayanan, Kim & Makino, 2003). The fundamental unit for the formation of coronaviruses (CoV) nucleocapsids is a dimeric structure composed of NP (Fan et al., 2005), with the C-terminal dimerization domain (CTD) of the NP providing the ability to dimerize (Surjit et al., 2004). A study found that structural analysis shows the NP is present not only in the helical nucleocapsid but also in the inner spherical or icosahedral core (Risco et al., 1996). RNA, the CTD of the M protein, and the NP are only a few of the vital elements that are located inside the virus’s internal core and work together to sustain healthy viral functioning. Most of the core–shell, or M protein, binds to the NP by an ionic interaction with a specific 16 amino acid region (aa 237–252) on its CTD. This enables the exact assembly of the genetic material within the growing viral particle (Risco et al., 1996; Escors et al., 2001; Kuo & Masters, 2002). Consequently. The NP is necessary for the CoV virion to assemble because of its interactions with the M protein, the genomic RNA, and other NP.

Viral assembly

The creation of viral particles is an important stage in ensuring the virus’s successful reproduction cycle. Within the viral envelope of CoV virions, there is a viral nucleocapsid consisting of gRNA and NP, along with three envelope proteins, envelope (E), membrane (M), and spike (S). The assembly process of CoV virions requires several steps, including the dimerization of CoV NP (He et al., 2004; Surjit et al., 2004; Yu et al., 2005) and its association with viral gRNA to form RNPs (Macnaughton, Davies & Nermut, 1978; Davies, Dourmashkin & Macnaughton, 1981; Baric et al., 1988; Risco et al., 1996; Nelson, Stohlman & Tahara, 2000; Chen et al., 2005), interactions between the four structural proteins, and the acquisition of a host membrane envelope from the site of budding. Through the use of endoplasmic reticulum (ER)-Golgi intermediate compartment membranes, nucleocapsid budding produces the lipid envelope of CoVs (De Haan et al., 1998; De Haan et al., 1999). The integration of the nucleocapsid in enveloped viruses is thought to depend on the connection between the nucleocapsid and envelope proteins (Suomalainen, Liljeström & Garoff, 1992), and this has been observed in alphaviruses since such interactions between proteins are essential for viral assembly (Suomalainen, Liljeström & Garoff, 1992; Lopez et al., 1994).

In CoV virions, the N and M proteins are the most significant structural proteins (Sturman, Holmes & Behnke, 1980; Narayanan et al., 2000). The M protein’s three transmembrane domains anchor it to the viral envelope, while its sizable carboxy-terminal tail in the virion interior engages with the nucleocapsid (Narayanan et al., 2000). Positive-strand gRNA and mRNA 1 in the nucleocapsid are helically enveloped by NPs. The N-M protein interaction at the C-terminus seems unique to CoVs.

Replication of genomic mRNA and synthesis of gRNA

In order to obtain a successful viral replication, the degradation of the virus genome via RNA interference as an inherent antiviral immune defense by the host should be inhibited (Mu et al., 2020; Bai et al., 2021; Wu et al., 2023). NP functions as an RNA interference inhibitor and it mediates the synthesis of gRNA (Cong et al., 2020; Cai et al., 2021; Xu et al., 2021). The NP has the ability to attach to both the complete gRNA and all sgRNAs, with a stronger attraction towards the gRNA (Cologna & Hogue, 1998). The gRNA serves as both a blueprint for the viral RNA-dependent RNA polymerase and a messenger for translation (Hurst et al., 2010). In infection, gRNA is transcribed into negative-stranded RNA, then full-length gRNA that attaches to polysomes (Brayton et al., 1982) and is observed in nucleocapsids (Spaan et al., 1981). A large number of studies have indicated that optimal CoV replication requires the presence of the NP (Yount, Curtis & Baric, 2000; Thiel et al., 2001; Casais et al., 2001; Yount et al., 2002; Yount et al., 2003; Almazán, Galán & Enjuanes, 2004; Schelle et al., 2005; Zúñiga et al., 2010). The implication that the N protein plays a role in an early stage of RNA synthesis is supported by two observations: first, the colocalization of mouse hepatitis virus (MHV) and SARS-CoV NPs with replicase components inside cells at the beginning of infection (Denison et al., 1999; Van Der Meer et al., 1999; Sims, Ostermann & Denison, 2000; Stertz et al., 2007), and second, the dependence of gRNA infection stimulation on NP translation (Masters et al., 1994; Hurst et al., 2010).

It has been shown that the regulation of phosphorylation of the SR-rich region (serine/arginine rich) of NP could affect viral replication and transcription (Bouhaddou et al., 2020; Liu et al., 2021; Carlson et al., 2022; Cheng et al., 2023). A clearer role for NP in gRNA synthesis was established when an interaction between MHV NP’s SR region and nsp3 subunit’s amino terminal segment was found (Hurst et al., 2010; Keane & Giedroc, 2013). The interaction between N and nsp3 has been precisely located in the ubiquitin-like domain (Ubl1) of nsp3, a crucial domain for the virus (Hurst, Koetzner & Masters, 2013). Furthermore, it has been established that the N-nsp3 interaction is essential for the NP to effectively increase the infectivity of gRNA (Hurst et al., 2010). It has been put forth that NP links gRNA to replicase via nsp3 interaction for initiation complex development at the genome’s 3′  end (Hurst et al., 2010; Hurst, Koetzner & Masters, 2013). Given that negative-strand synthesis initiation is the initial step in both genomic replication and transcription (Sawicki, Sawicki & Siddell, 2007; Züst et al., 2008), the N-nsp3 interaction is likely significant in both RNA-synthetic processes (Hurst et al., 2010). The involvement of the NP in MHV gRNA synthesis may not be restricted to the early phases of infection, as it has been proposed that the NP supports ongoing transcription for the duration of the infection (Graham & Denison, 2006; Zúñiga et al., 2010). The contribution of the CoV NP to RNA synthesis is still disputed, as it’s not indispensable for transmissible gastroenteritis coronavirus RNA replication, but it plays a significant role in efficient transcription (Zúñiga et al., 2010).

Chaperone activity

RNA chaperone proteins provide assistance in ensuring the correct folding of nucleic acid molecules (Herschlag, 1995; Cristofari & Darlix, 2002). It has been suggested that chaperone activity is a common feature of all CoV NPs, as evidenced by the demonstration of this activity in transmissible gastroenteritis coronavirus and SARS-CoV NPs (Zúñiga et al., 2007). Chaperone activity and in vitro template switching facilitation are proposed for amino acids 117-268 of transmissible gastroenteritis coronavirus NP’s linker region domain (Zúñiga et al., 2007; Zúñiga et al., 2010). It has been demonstrated that NP increases RNA synthesis; this effect may be attributed to the protein’s ability to facilitate template switching, which is necessary for effective transcription.

Cell cycle regulation

The exploitation of host cell machinery through the deregulation of the cell cycle is a frequently used tactic by many RNA and DNA viruses to create a more favorable environment for their survival. The MHV nonstructural protein, p28, is known to cause G0/G1 cell cycle arrest by preventing the hyperphosphorylation of retinoblastoma protein (Rb), a crucial step for the progression of the cell cycle from late G1 into the S phase (Ikeda et al., 1998; Chen & Makino, 2004). A model suggests that the expression of cytoplasmic p28 leads to the stabilization of cellular tumor antigen p53, and the subsequent accumulation of p53 results in the transcriptional upregulation of p21, a cyclin-dependent kinase (CDK) inhibitor. This, in turn, suppresses cyclin E/CDK 2 activities and reduces G1 cyclin-CDK complexes and CDK activities, ultimately inhibiting Rb hyperphosphorylation (Chen & Makino, 2004). The NP of SARS-CoV regulates cyclin-CDK activity to modulate the host cell cycle, resulting in the halting of S-phase progression. The NP, which possesses a signature cyclin box-binding region (RXL motif), is capable of being phosphorylated by CDK and thus serves as a substrate for the cyclin-CDK complex (Surjit et al., 2005). The NP has been demonstrated to inhibit S-phase cyclins CDK4 and, to a lesser degree, CDK6 (Surjit et al., 2005). Kinase inhibition causes Rb to remain hypophosphorylated, preventing E2F1 release and halting S phase gene transcription (Ikeda et al., 1998). In a manner similar to MHV p28, the NP of SARS-CoV also inhibits CDK2 activity, which results in the blockage of both G1 and S phase cyclins. This means that the NP doubly ensures the prevention of S phase progression (Surjit et al., 2005). NP inhibits CDK activity, leading to Rb phosphorylation inhibition, independently of CDK inhibitors (CDKIs) like p21. NP is suggested to mimic CDKIs by competitively inhibiting CDK4 and CDK2 (Surjit et al., 2005). It has been suggested that the blocking of the S phase by p28 (MHV) and NP (SARS) allows these viruses sufficient time to use the cellular raw materials synthesized prior to the S phase for genome replication and assembly and budding of progeny particles (Surjit et al., 2006). The results obtained show that different proteins belonging to the CoV family can reduce the regulation of the cell cycle; in the case of the SARS-NP, multiple pathways can be used to do this.

SARS-CoV-2 NP: Expression and Immunity

NP establishes a connection with viral RNA via electrostatic interactions, which results in the creation of virion particles consisting of cytoplasmic helical ribonucleoprotein complex (Di et al., 2021). These nucleocapsids subsequently associate with M, promoting the emergence of the virus into early secretory compartments. N, being the most profusely expressed protein of SARS-CoV-2, triggers potent Ab and TCD8+ immune responses (Long et al., 2020; Sariol & Perlman, 2020; López-Muñoz et al., 2022).

The phenomenon of viral RNA- and DNA-binding proteins being expressed on the cell surface can be traced back to the initial detection of polyclonal Ab of retrovirus gag and polyoma virus T antigen in the 1970s (Deppert & Henning, 1980). Contrary to the widespread assumption that CoV N is solely found in the cytoplasm, it’s often observed that the cell surface expression of RNA virus’s N is more common (López-Muñoz et al., 2022). Preliminary experiments using monoclonal Abs (mAbs) have documented the surface expression of influenza A and vesicular stomatitis virus N (Yewdell, Frank & Gerhard, 1981; Yewdell et al., 1986). Influenza neuraminidase (N) protein, a significant component of the influenza virus, is susceptible to two specific types of lysis: Ab complement–mediated cell lysis (Yewdell, Frank & Gerhard, 1981) and Ab-redirected T cell lysis (Staerz, Yewdell & Bevan, 1987). These processes contribute to the destruction of infected cells. Additionally, in the context of murine models (LaMere et al., 2011), Influenza N is specifically targeted by protective Abs, which play a crucial role in the immune response against the virus. This is relevant to SARS-CoV-2 because similar mechanisms of immune response and cell surface expression of the nucleocapsid protein have been observed. Research on the pure SARS-CoV-2 NP recombinant from bacterial cells demonstrates that it triggers an immune response in mouse and human mAbs (Djukic et al., 2021; Di et al., 2021; Li & Li, 2021).

Studies on viral infections in humans have shown that N and N-like RNA genome binding proteins are expressed on the surface of infected cells. Included in this group of viruses are the lymphocytic choriomeningitis virus (Straub et al., 2013), the HIV virus (Ikuta et al., 1989), measles (Marie et al., 2004), and the respiratory syncytial virus (Céspedes et al., 2014). This can be also applied to the NP from SARS-CoV-2 (López-Muñoz et al., 2022), which can regulate chemokine synthesis (López-Muñoz, Santos & Yewdell, 2023). Research findings have revealed that the immune system response can be negatively impacted by NP expression on the cell surface. In particular, it was discovered to block immune synapses located on T cells, which are essential for effective immune responses. In addition, it can suppress Il-12 release, an important cytokine engaged in immunological signaling (Marie et al., 2004; Céspedes et al., 2014).

Studies conducted on the NP of SARS-CoV-2 have demonstrated that it significantly affects the outcomes of both innate and adaptive immunity. It inhibits the expression of type I and type III interferons. Specifically, it inhibits the IFN-β (interferon beta), NF-κB, and IFN-λ1 (interferon lambda 1) promoter activities (Chang et al., 2020; Li et al., 2020), and inhibits the interaction between retinoic acid-inducible gene I (RIG-I) and tripartite motif protein 25 (TRIM25) (Zhao et al., 2021a; Mohd Faizal et al., 2023). Additionally, NP increases the viability of infected cells by demonstrating an anti-apoptotic action via modifying the BAX (apoptosis regulator BAX) and BCL-2 (apoptosis regulator BCL-2) genes (Edalat et al., 2024).

The NP has the ability to control nuclear translocation and phosphorylation of IRF3 (interferon regulatory factor 3), STAT1 (signal transducer and activator of transcription 1), and STAT2 (Zhao et al., 2021a). Moreover, it has been shown that NP has a high-affinity interaction with more than 10 human chemokines, including CXCL12β (C-X-C motif chemokine, 12 also known as stromal cell-derived factor 1, SDF-1). This is an important mechanism by which the chemotaxis of leukocytes, normally performed by CXCL12β is inhibited (López-Muñoz, Santos & Yewdell, 2023). It has also been shown that SARS-CoV-2 NP leads to the overexpression of proinflammatory cytokines in general, including IL-6 (interleukin 6), IL-12, IL-1β, and TNF-α (tumor necrosis factor alpha) (Edalat et al., 2024), as well as the triggering of cytokine storm in lung epithelial cells specifically (Wang et al., 2024). One of the ways a proinflammatory response can happen is through the interaction with 14-3-3 human isoforms (all seven of them); a group of proteins important for different types of signaling in cells, including the modulation of the inflammatory response (Munier, Ottmann & Perry, 2021; Tugaeva et al., 2021).

The Dormant Endothelial Layer

A monolayer of cobblestone-shaped endothelial cells that are placed at the inner lining of blood vessels and act as a link between the circulatory system and organ-specific tissues forms the endothelium (Pober & Sessa, 2007; Perico, Benigni & Remuzzi, 2024). To comprehend the role of the endothelium in the vascular complications of COVID-19, it’s essential to first understand its functions in normal physiology.

The endothelium is <0.2 µm thick and it is covering the central parts of the immune and vascular system (Wolinsky, 1980; Gomez-Salinero & Rafii, 2018). The structure of the endothelial cells is different based on the tissue of the origin (Gomez-Salinero & Rafii, 2018) and their function mostly includes maintenance of vascular homeostasis (Murakami et al., 2008), the control of vascular tone and blood flow (D’Alessio, 2023), and serving as an anticoagulant surface and maintains barrier integrity (Van Hinsbergh, 2012). They also play a key role in immune surveillance, contributing to both adaptive and innate immunity (Mai et al., 2013; Shao et al., 2020). Specific inherent cytoprotective mechanisms and unique cellular features ensure the maintenance of endothelial dormancy.

A balanced condition of the environment in contact with endothelium leads to anticoagulant and non-thrombogenic cells (Félétou, 2011). However, any changes that cause activation of endothelial cells and production of macromolecules by them could potentially promote thrombus formation (Félétou, 2011). Endothelial activation, also known as endotheliopathy, has been reported as a consequence of the presence of SARS-CoV-2 particles (Teuwen et al., 2020; Goshua et al., 2020; Zhang et al., 2020; Nassar et al., 2021; Otifi & Adiga, 2022; Perico, Benigni & Remuzzi, 2024). Multiple evidence has shown that patients who were diagnosed with this dysfunction as a result of SARS-CoV-2 infection, developed endothelitis, hypercoagulation, and hypofibrinolysis (Valencia et al., 2024). Furthermore, the link between the severity of CoV-19 infection, disease mortality, and contributing to long-COVID syndrome and post-COVID sequela to endotheliopathy makes the understanding of this topic more crucial (Teuwen et al., 2020; Otifi & Adiga, 2022; Perico, Benigni & Remuzzi, 2024; Valencia et al., 2024).

Hyperinflammation and Thrombosis: The Link Between COVID-19, Long COVID, and the Nucleocapsid Protein

Coronaviruses from previous epidemics, specifically SARS-CoV and MERS-CoV, have been implicated in increasing the propensity for thrombosis (Giannis, Ziogas & Gianni, 2020). Coagulopathy is a significant clinical symptom of COVID-19. The development of this complication involves various mechanisms such as damage to the endothelium, an excessive inflammatory response and cytokine storm, activation of the complement system, mononuclear phagocytes, formation of neutrophil extracellular traps (NETs), and tissue hypoxia (McFadyen, Stevens & Peter, 2020; Abou-Ismail et al., 2020; Goswami et al., 2021).

The cytokine profile of patients with severe COVID-19 shows a rise in the production of TNF, IL-6, IL-7, and inflammatory chemokines including CCL3, CCL2, and soluble IL-2 receptors. These findings parallel those found in cytokine release syndrome (CRS), such as macrophage activation syndrome. Thrombosis can result from an overproduction of cytokines, which activate monocytes, neutrophils, and the endothelium, thereby promoting a prothrombotic state (Abou-Ismail et al., 2020). During proinflammatory conditions, IL-1α, expressed by activated platelets, endothelial cells, and monocytes, acts as a connector between the coagulation process and inflammation. Moreover, IL-1α contributes to thrombosis by extending clot lysis time, enhancing platelet function, and stimulating endothelial cells. The secretion of IL-1α by epithelial cells triggers the detection by inflammatory myeloid cells and inflammasome activation, leading to an intensified inflammatory cascade (Di Salvo et al., 2021). At the same time, endothelial cells expressing IL-1α trigger the attraction of granulocytes and thrombosis (Savla, Prabhavalkar & Bhatt, 2021).

From a mechanistic standpoint, it’s postulated that the NP of SARS-CoV-2 has the ability to bind to MASP-2 (mannan-binding lectin serine protease 2), a crucial component in the complement system (Gao et al., 2022). This interaction is believed to trigger the activation of the complement system, a key player in the body’s immune response, thereby intensifying the injury to tissues. The subsequent activation of the complement pathway is associated with an increased production of various endothelial cytokines, including but not limited to IL-6, RANTES (T-cell-specific protein RANTES, also known as C-C motif chemokine 5, CCL5), IL-1, IL-8, and MCP-1 (monocyte chemoattractant protein 1, also known as C-C motif chemokine 2, CCL2). The body’s inflammatory response is significantly influenced by these cytokines. In addition, the activation also leads to an upregulation in the expression of important endothelial adhesion molecules, specifically P-selectin and the von Willebrand factor, which are vital for cellular adhesion and thrombus formation (Foreman et al., 1994; Foley & Conway, 2016).

Our previous review hypothesized that the SARS-CoV-2 NP might contribute to long COVID through mechanisms such as liquid-liquid phase separation (Eltayeb et al., 2024), potentially leading to persistent inflammation and thrombosis. This hypothesis is supported by the multifunctional nature of the NP, which interacts with various host proteins and modulates immune responses (Katsoularis et al., 2022) focused on respiratory endothelial activation rather than thrombosis (Chen et al., 2022). They demonstrated that the NP of SARS-CoV-2 activates endothelial cells via the TLR2/NF-κB and MAPK signaling pathways. While this study did not directly conclude on thrombosis, endothelial activation is a critical step in the pathway leading to thrombotic events. Similarly, Qian et al., (2021) provided insights into the endothelial activation induced by the SARS-CoV-2 NP. They found that the NP significantly activates human endothelial cells through TLR2-mediated signaling pathways. Although this study did not demonstrate thrombosis in human or animal models, endothelial activation is a known precursor to thrombosis, supporting the potential link between the NP and thrombotic events.

Additional evidence further supports the association between COVID-19 and thrombotic complications. A recent study reported a significant increase in the risk of deep vein thrombosis and pulmonary embolism following SARS-CoV-2 infection (Othman et al., 2024). This study underscores the heightened thrombotic risk associated with COVID-19, which may extend to long COVID. Furthermore, research has shown that long COVID is associated with persistent coagulation abnormalities, including thrombotic complications (Jenner et al., 2021). This highlights the importance of understanding the mechanisms by which SARS-CoV-2, particularly the NP, may contribute to these long-term effects. By integrating these findings, we provide a comprehensive overview of the potential mechanisms linking the SARS-CoV-2 NP to thrombosis and long COVID. Further research is needed to elucidate these mechanisms and confirm the direct role of the NP in thrombotic events.

In addition, another study on the nucleocapsid protein of SARS-CoV-2 demonstrated that it provokes hyperinflammation through the accumulation of intracellular Cl- in the respiratory epithelium, mediated by protein-protein interactions. The researchers delved into the mechanism by which the NP of SARS-CoV-2 instigates excessive inflammatory responses in respiratory epithelial cells. Their findings indicated that the NP interacts with SMAD3 (mothers against decapentaplegic homolog 3) and suppresses the expression of CFTR (cystic fibrosis transmembrane conductance regulator), leading to an elevation in intracellular Cl− concentration. This, in turn, results in the phosphorylation of SGK1 (serum/glucocorticoid-regulated kinase 1, also known as serine/threonine-protein kinase SGK1) and triggers subsequent inflammatory responses (Chen et al., 2022).

Research conducted by Qian et al. (2021) determined several pathways related to NP and endothelial cells. The MAPK and NF-κB signaling pathways regulate the expression of ICAM-1 and VCAM-1 (Fusté et al., 2004). By performing Western blott analysis, they determined that NP treatment induced the phosphorylation of IκB kinases (IKKs), p65, IκBα, JNK (c-Jun N-terminal kinase 1, also known as mitogen-activated protein kinase 8), and p38 and caused IκBα degradation suggesting that NP activated JNK, p38, and NF-κB signal pathways in human endothelial cells (Qian et al., 2021). The same group determined NP-induced endothelial cell activation via the TLR2-mediated signaling pathway. It was shown that internalization of the NP is not required for endothelium activation, as indicated by the absence of impact from endocytosis inhibitors and NP overexpression within cells. The NP seems to interact with a cell surface receptor, most likely TLR2, as TLR2 antagonists, but not TLR4 or IL-1R, greatly reduced NP-induced ICAM-1 and VCAM-1 expression. Moreover, the research done by the same working group (Qian et al., 2021) observed that SARS-CoV-2’s NP, but not those of other coronaviruses, significantly increased endothelial cell activation. Despite the substantial sequence similarity between SARS-CoV-2, SARS-CoV, and MERS-CoV, only SARS-CoV-2 causes major vascular damage and thrombosis. This difference is due to SARS-CoV-2’s NP, which, unlike other viruses’ NPs, promotes endothelium activation. This result explains the significant vascular problems reported in COVID-19 patients (Qian et al., 2021).

While a significant proportion of individuals recover from the acute phase of COVID-19, a subset continues to experience symptoms for an extended period, often weeks or months after the initial infection. The enduring impacts of hyperinflammation and thrombosis, two critical aspects of severe COVID-19, are likely implicated in this syndrome. A particular article delves deeper into the potential role of the SARS-CoV-2 nucleocapsid protein in the development of long COVID, a condition where individuals continue to experience symptoms for more than 12 weeks after being infected with the virus. The authors highlight the fundamental studies conducted on the nucleocapsid protein, particularly its capacity to go through a mechanism known as liquid-liquid phase separation. They suggested that this unique property may contribute to the persistent inflammation and bothersome symptoms experienced by individuals with long COVID (Eltayeb et al., 2024).

Numerous research efforts have indicated that a lack of or minimal production of SARS-CoV-2 antibodies, along with other inadequate immune responses during the acute phase of COVID-19, can forecast the occurrence of long COVID after a period of 6–7 months in patients, irrespective of whether they were hospitalized or not (Augustin et al., 2021; Ballering et al., 2022). The inadequate immune responses encompass a diminished baseline level of IgG (Augustin et al., 2021), spike-specific memory B cells and reduced quantities of receptor-binding domain (García-Abellán et al., 2021), low peak values of spike-specific IgG, and decreased levels of nucleocapsid IgG (García-Abellán et al., 2021). In patients with severe long COVID, a recent preprint emphasized the low or absent responses of CD8+ T cells and CD4+ T cells (Talla et al., 2021). Furthermore, researchers found lower numbers of CD8+ T cells expressing CD107a, a crucial marker of cytotoxic T cell activity, in another extensive investigation. Concurrently, there was a decline in the quantity of CD8+ T cells that produced interferon-γ specific to nucleocapsid. When these patients were contrasted with a control group of people who had contracted the virus but did not suffer the long-term consequences of long COVID, the results were notably clear in those suffering from severe long COVID (Peluso et al., 2021). The presence of high autoantibody levels in long COVID has been found to negatively correlate with the levels of protective COVID-19 antibodies. This indicates that patients with elevated autoantibody levels could be more susceptible to breakthrough infections (Su et al., 2022).

Conclusions

This review has illustrated the critical role of the SARS-CoV-2 nucleocapsid (N) protein in COVID-19, based on an analysis of 149 selected studies. The NP is prevalent in coronaviruses and plays an essential role in viral replication and assembly. While NPs from SARS-CoV and MERS-CoV have been shown to increase inflammation and cause acute lung injury, the specific effects of the SARS-CoV-2 NP on host cells are still being explored. It has been recognized that the SARS-CoV-2 NP acts as a pathogen-associated molecular pattern (PAMP) that binds to Toll-like receptor 2 (TLR2) and activates the NF-κB and MAPK signaling pathways. SARS-CoV-2 infection activates CK2 and p38 MAPK, raising blood levels of soluble ICAM-1 and VCAM-1, especially in severe cases. This underscores the significance of the NP in endothelial activation, leading to extensive endothelial dysfunction and multiorgan damage reported in severe COVID-19 patients.

Conclusively, the nucleocapsid protein of SARS-CoV-2 has been pinpointed as a crucial element in the evolution of hyperinflammation and thrombosis, two significant determinants in the severity and persistent effects of COVID-19. Its potential to bind to MASP-2 and initiate liquid-liquid phase separation may be associated with the ongoing symptoms endured by those affected by long COVID. Further investigation and insight into the mechanisms implicated in this protein’s interaction with the immune system could result in improved therapeutic approaches and management of both the immediate and long-term consequences of COVID-19. The research on the SARS-CoV-2 NP is marked by both controversies and promising new directions. Conflicting evidence about the effects of NP phosphorylation and mutations highlights the complexity of its role in the viral lifecycle. However, emerging perspectives on targeting the NP for diagnostics, vaccines, and therapeutics offer hope for more effective management of COVID-19 and its long-term effects. The highly conserved nature of the NP makes it a promising target for diagnostic tests and vaccines, potentially improving sensitivity and effectiveness. Furthermore, understanding how the NP modulates the immune response, such as its ability to inhibit the NF-κB pathway, provides new insights into potential therapeutic strategies.

Overall, the SARS-CoV-2 NP represents a promising focal point for future research and therapeutic development. The ongoing efforts in this field are a testament to the resilience and dedication of the scientific community. By focusing on the NP, researchers and healthcare professionals can develop more effective diagnostic tools, vaccines, and therapeutic strategies. This will not only enhance our ability to manage and treat COVID-19 but also address the persistent challenges posed by long COVID. The advancements in understanding and targeting the NP hold the promise of significantly improving patient outcomes and public health worldwide. The scientific community’s commitment to unraveling the complexities of the NP will undoubtedly lead to breakthroughs that will benefit millions of people affected by COVID-19.

Summary and Future Directions

This review has delved into the pivotal role of the SARS-CoV-2 NP in the pathogenesis of COVID-19, emphasizing its involvement in hyperinflammation and thrombosis, which are critical factors in the disease’s severity and long-term effects. The NP’s interactions with host cell components, such as Toll-like receptor 2 (TLR2) and various signaling pathways, including NF-κB and MAPK, underscore its significant impact on endothelial dysfunction and immune response modulation. Key takeaways from this review include the NP’s essential functions in viral replication and assembly, its ability to trigger hyperinflammation through interactions with host proteins, and its role in promoting thrombosis and endothelial dysfunction, which contribute to COVID-19-associated coagulopathy and long COVID symptoms. Understanding these mechanisms opens avenues for developing targeted therapies to mitigate both acute and chronic effects of COVID-19.

Despite significant advancements in understanding the SARS-CoV-2 NP, several knowledge gaps remain that warrant further investigation. Addressing these gaps is crucial for developing effective therapeutic strategies and improving our overall understanding of COVID-19 and its long-term effects. While it is known that the NP can modulate immune responses, the precise mechanisms by which it interacts with host immune pathways, such as NF-κB and MAPK, are not fully understood. Detailed studies are needed to elucidate how the NP influences these signaling pathways and the subsequent effects on immune cell activation and cytokine production. The exact role of the NP in the development and persistence of long COVID symptoms remains unclear. Longitudinal studies focusing on patients with long COVID are essential to determine the NP’s contribution to persistent symptoms. Investigating the protein’s interactions with host cells over time could provide insights into chronic inflammation and immune dysregulation associated with long COVID.

Mutations in the NP, particularly those affecting phosphorylation sites, have been linked to changes in viral replication and assembly. However, the broader implications of these mutations on viral fitness, immune evasion, and disease severity are not fully understood. Comparative studies of different SARS-CoV-2 variants with specific NP mutations can help clarify how these changes affect the virus’s behavior and interaction with the host immune system. Although the NP is a promising target for antiviral therapies, there is limited information on the efficacy and safety of potential inhibitors. High-throughput screening of small molecules and peptides that can inhibit NP functions should be prioritized. Structural studies using techniques like cryo-electron microscopy can aid in the rational design of inhibitors. Additionally, preclinical and clinical trials are necessary to evaluate the therapeutic potential of these inhibitors.

The interplay between the NP and the host immune response is complex, and its implications for therapeutic development are not fully explored. Investigating the immune response to the NP, including the generation of neutralizing antibodies and T-cell responses, can inform vaccine design and immunotherapies. Studies should focus on how to enhance protective immunity while minimizing potential adverse effects, such as immune enhancement or autoimmunity.

In conclusion, the SARS-CoV-2 NP represents a promising focal point for future research and therapeutic development. The ongoing efforts in this field are a testament to the resilience and dedication of the scientific community. By focusing on the NP, researchers and healthcare professionals can develop more effective diagnostic tools, vaccines, and therapeutic strategies. This will not only enhance our ability to manage and treat COVID-19 but also address the persistent challenges posed by long COVID. The advancements in understanding and targeting the NP hold the promise of significantly improving patient outcomes and public health worldwide. The scientific community’s commitment to unraveling the complexities of the NP will undoubtedly lead to breakthroughs that will benefit millions of people affected by COVID-19.

Additional Information and Declarations

Competing Interests

Author Contributions

Data Availability

Altijana Hromić-Jahjefendić is an Academic Editor for PeerJ. Vladimir N. Uversky is an Academic Editor and Section Editor for PeerJ. The other authors declare that they have no competing interests.

Ahmed Eltayeb conceived and designed the experiments, performed the experiments, analyzed the data, authored or reviewed drafts of the article, and approved the final draft.

Muhamed Adilović performed the experiments, analyzed the data, authored or reviewed drafts of the article, and approved the final draft.

Maryam Golzardi performed the experiments, analyzed the data, authored or reviewed drafts of the article, and approved the final draft.

Altijana Hromić-Jahjefendić conceived and designed the experiments, performed the experiments, analyzed the data, authored or reviewed drafts of the article, and approved the final draft.

Alberto Rubio-Casillas performed the experiments, analyzed the data, authored or reviewed drafts of the article, and approved the final draft.

Vladimir N. Uversky conceived and designed the experiments, performed the experiments, analyzed the data, prepared figures and/or tables, authored or reviewed drafts of the article, and approved the final draft.

Elrashdy M. Redwan conceived and designed the experiments, performed the experiments, analyzed the data, authored or reviewed drafts of the article, and approved the final draft.

The following information was supplied regarding data availability:

This is a literature review.

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
