# Peer review of "Intrinsic factors behind long COVID: exploring the role of nucleocapsid protein in thrombosis"

_PeerJ, doi:10.7717/peerj.19429_

## Round 0.1 · original submission · Major Revisions

Four experts in the field reviewed your manuscript. As you can see from their comments below, all of them give substantial points to be revised. Particularly, I think the manuscript should be rewritten with a more precise logical flow, and the relationship between COVID-19 and thrombosis should be treated more carefully. Please read their comments carefully and revise the manuscript accordingly.

·

Basic reporting

The actual objective of the review is a bit unclear as is stated differently several times.
i.e. ‘This review illuminates our understanding of the expression of SARS-CoV-2 to N protein and its consequences’.
‘This review paper aims to investigate the role of the NP in platelet function in COVID-19 and its post-acute sequealae and to identify areas for future research’.
‘The article is designed for researchers doctors and healthcare professionals interested in the investigation and treatment of COVID-19 and its post-acute complications’
‘It seeks to get a thorough knowledge of the role of the SARS-CoV-2 nucleocapsid protein in viral replication immune response regulation and COVID-19 pathogenesis.’

The Abstract text could do with a restructure and some more detail to make clear and more compelling. Currently is more like a series of somewhat disconnected sentences. Key concepts are not really introduced i.e. that thrombosis occurs during severe COVID-19 infection. Also, the Review title is ‘Intrinsic factors behind long COVID’ but this condition is not mentioned at all.

Similarly in the Introduction, is overall fairly strong, but some of the sentences are out of place resulting in a bit of a jumbled structure/flow. I.e. starts by introducing SARS-Cov_2 and COVID-19 disease and range of symptoms. Explains viral entry via ACE2, and viral clearance by many but also long COVID. Then brings up viral transmission routes, then introduces the nucleocapsid protein, returns to explaining entry via ACE2, then brings up transmissibility aspects again. Then outlines possible symptoms in more detail.

Experimental design

Under Survey methodology, I think is a bit of an exaggeration to state that the authors can ‘…guarantee complete an unbiased coverage of the SARS-Cov-2 literature’, and I also do not consider this Review to be a ‘systematic review’. The Survey methodology described seems just to be the standard approach to preparing a comprehensive, thorough Literature review. Whilst yes incorporated an ‘unbiased’ initial search, it then involved lots of subjective reading and drawing out of information to synthesise together. The purpose of this manuscript is to give some overall information and insights (into SARs-Cov-2, NP, long covid, etc) rather than answer a specific research question. I would personally just cut this Section entirely.

More explanation of Fig1A is required. I.e. define what a disordered region/protein is. In the text state how you have generated / sourced the structures presented. Similarly for D, how was this interaction map generated.

There is a lack of citations given for some statements.

Just to illustrate the repetitiousness,
Start of Section 3, Line 97: ‘The genetic code of SARS-Cov-2 is composed of around 30,000 nucleotides that specify the production of four key structural proteins the S,E,M and N proteins’ We are introduced to the structural proteins of SARS-Cov-2, in particular NP and its function.
Start of Section 4, Line 139: ‘SARS-Cov-2 viral RNA synthesis and packaging is encoded by its genomic RNA (gRNA) that is length around 30kb long. This gRNA includes structural proteins such as S spike protein memb rane protein E envelope protein N nucleocapsid protein and more than 20 non structural proteins (nsp).’ We are again introduced to the structural proteins of SARS-Cov-2, including NP, with its function in more detail.
Section 4.1, Line 172: ‘within the viral envelope of covariance there was a viral nucleocapsid consisting of gRNA and protein along with three envelope proteins M, E and S’
Start of Section 5, Line 268: ‘the Covs are characterised by the encoding of four essential structural proteins the S, M and E proteins are prominently located on the virus’s envelope surface’

Validity of the findings

This review has some fundamental flaws, in particular related to structure. There is a fair bit of repetition, with many sentences seeming out of order / disconnected. As a result, ideas are not conveyed clearly. There is also an excess of information not related to the main purpose.

Additional comments

Minor typographic issues. Thorough edit recommended.
‘Signalosome’ I think is too specific a term to drop in with no context/explanation.
It is confusing to jump around between the two abbreviations of N protein and NP.

·

Basic reporting

Suggest to:
- Simplify the language to make it more academic and factual.
- Avoid overly complex or non-academic phrasing. Break down lengthy sentences for better readability.
- Ensure consistency in terminology and avoid unnecessary repetition.
- Include specific references to support key claims throughout the review.
- Ensure uniformity in citation format and ensure that recent and updated literature (including studies from 2024) are referenced to maintain relevance.
-Rewrite "COVID-19, one of the three severe coronavirus epidemics in the past two decades, is a
major global health concern".
-Rewrite "It resulted in the coronavirus disease 2019 pandemic (COVID-19), an acute respiratory infection that posed a worldwide health threat, Please use SARS-CoV-2 for the virus and Covid_19 for the disease only.
-Abbreviate key immune response components and pathway-related proteins like CK2, p38 MAPK, etc., after their first mention for readability and consistency.
-Align the abstract with the title by including a mention of the nucleocapsid protein and its role in thrombosis toward the end of the abstract. This will improve thematic consistency.

Experimental design

Following comments to improve the submitted article:
- Please ensure the abstract must align with the title and body of the manuscript.
- Suggesting authors present their own viewpoint, especially in the discussion and conclusion sections.
- Their perspective should be backed by evidence but framed as an insightful contribution to the field.
- Address instances where the influenza virus is mentioned (line 280) and clarify its relevance to SARS-CoV-2 to avoid confusion.
- Ensure that all external references are directly tied to the manuscript's theme to avoid unnecessary detours.
- There is a lack of consistency in thematic focus.

Validity of the findings

Following comments to improve the submitted article:
- Add a summary section at the end of the review that highlights key takeaways, future directions, and the impact of this review on the field.

- Highlight any controversies, conflicting evidence, or emerging perspectives in SARS-CoV-2 N protein research focus to long covid.

Additional comments

Figure Clarity:
- Improve the resolution and clarity of Figure 1 to make it visually appealing and easy to interpret.
-- Add a stronger concluding section with supportive and affirmative remarks.
- Include a critical commentary on knowledge gaps and potential areas for future research, especially regarding the role of immunity in therapeutic development.
- Authors should provide their perspective on the reviewed literature, adding depth to the conclusion.

Reviewer 3 ·

Basic reporting

- The study is broad and a good survey article. The exploration of the connection between thrombosis and long COVID outcomes is novel.
- The paper needs to identify gaps and future directions for thoroughness.
- The introduction adequately addresses the subject

Experimental design

- The study methodology section could provide number of studies selected, and inclusion and exclusion criteria. This also needs to be discussed in a Discussion section
- The paper needs to mention in details the figures generated and data underlying them (especially figure 1A)

Validity of the findings

- Liquid-Liquid separation is referenced only from a single study separately. This needs to be discussed. Also the endothelial dysfunction is from a single study.
- What is the data underlying the figures? Please mention it thoroughly in the paper
- All figures needs to mention the source of the data (ex: how were they generated); including the Electron microscopy analysis mentioned. References needs to be provided as well.
- The conclusion needs to address the gaps and tie back to introduction
- Are there any alternative hypotheses and contradictory findings while researching the articles? this could go in Discussion section or in the individual sections

Additional comments

- A Discussion section is needed to discuss any bias, exclusion criteria, gaps, clinical implications

Annotated reviews are not available for download in order to protect the identity of reviewers who chose to remain anonymous.

Reviewer 4 ·

Basic reporting

The review submitted by Eltayeb et al. reviewed the role of the nucleocapsid(N) in thrombosis. The title is "Intrinsic factors behind long COVID: exploring the role of nucleocapsid protein in thrombosis". The review mostly described N protein's role in virus assembly and pathogenesis with little connection to thrombosis. The author also failed to develop a model that connects how N protein contributes to thrombosis or long COVID-19. The authors described different findings as reporting rather than connecting them to a common theme.

The introduction does not adequately introduce thrombosis and long COVID to the reader.. Rather, it's a general intro about how SARS-CoV-2 infection leads to COVID-19. A proper introduction of thrombosis and long COVID need to be addressed. The reader then can connect the symptoms and mechanisms to the N protein's role.

The review described mostly molecular biology of N protein in virus life cycle and its effect on cells. For thrombosis and long COVID, the author failed to show enough evidence that connect N proteins to thrombosis. The review describe very little to it's title, rather it describes general function of N protein of SARS-CoV-2 and related coronoviruses. To help the audience, the author can connect the evidence they saw in the literature that suggests the N protein's role in thrombosis and long COVID. The authors also need to investigate how much of the evidence clearly shows these connections. In the conclusion section, authors can describe the model they found from their literature review.

Experimental design

In this submitted review, a majority of claims are not related to SARS-CoV-2 rather SARS-CoV-1, MHV, Coronavirus A59, TGEV, IBV (Line 153-191, 197-209, 214-265, there are more). These claims are for N of other coronaviruses, even though in the methodology sections, the author attested,

"Our search method included important phrases such as "SARS-CoV-2," "COVID-19," "Nucleocapsid protein," "Platelet function," "Immunothrombosis," "Long COVID," and "Coagulopathy" utilizing Boolean operators. We considered peer-reviewed original research papers, reviews, and clinical trials published between December 2019 and the present, limiting our selection to English-language literature to guarantee accessibility and comprehensibility."

The claims mentioned above (in the review) are unrelated to SARS-CoV-2, and some were published in the 80s. Thus, the authors' findings are not entirely SARS-CoV-2 related.

The review article misses necessary citations in multiple places. Here are the claims need citations:
1. Line: 48-49, 59-61, 62-69, 371-372.

The review is not properly structured to follow easily. Each paragraph ends without connecting to the next- making it difficult to connect the authors' messages. Please make it coherent for the readers.

Validity of the findings

In the abstract, they described the general functions of Nucleocapsids like inflammation and virus life cycle, but failed to connect what role N protein plays in thrombosis. The only line they have for thrombosis is this "Over three years into the SARS-CoV-2 pandemic, it's evident that platelets contribute to the pathogenesis of COVID-19 and probably heighten the thrombosis risk, despite some contradictory and the exact mechanisms being unclear." The abstract does not connect with the title. The abstract needs to be corrected to reflect the title and the review.

The section (lines 349-435) where the author discussed Thrombosis and long COVID for evidence of thrombosis regarding SARS-CoV-2 are three articles mentioned by the authors. Interestingly, one article turned out to be the first author's review article (lines 407-417 titled: "Intrinsic factors behind long COVID: IV. Hypothetical roles of the SARS-CoV-2 nucleocapsid protein and its liquid-liquid phase separation." The abstract of this review discusses N protein function and the possibility of N protein's liquid-liquid phase separation causing long-term COVID. The other two articles- Chen et al., 2022 did not conclude on thrombosis but rather respiratory endothelial activation. On the other hand, Qian et al., 2021, did a protein expression analysis and linking to endothelial activation of N but none of these studies can confidently claims N protein in human or animal model can cause thrombosis.

Additional comments

The sentences need grammar check: Line 45: "the ACE2 protein endothelial cells" should be "the ACE2 protein of endothelial cells", 140: "This gRNA includes" should be "This gRNA encodes".

Need clarifications- the sentence is confusing: Lines 193-194.

Please use SARS-CoV-2 instead of COVID-19 in these sentences: Line 142: infected by COVID-19, 340: COVID-19 particles, 342-343: COVID-19 Infection,

---

## Round 0.2 · Minor Revisions

Of the four original reviewers, two agreed to review your revised manuscript. As you can see, both request additional minor comments. Please re-revise the manuscript accordingly.

·

Basic reporting

Minor comments -
Abbreviations introduced with no definition/explanation i.e. MASP-2 (Abstract), signalosome (Line 44), D-dimer (Line 62),
I think is a bit odd to say (Line 44) that ‘SARS-CoV-2 has been the topic of research on >>’ and then list 3 things. Would make sense to say something to the effect that there has been a huge array of research related to SARS-Cov-2 covering a all manner of topic areas, including >>.
Line 47, ‘.COVID-19, which can manifest as anything from no symptoms to severe pneumonia.’ I think could make sense to capture that the final /most severe outcome of COVID-19 is respiratory failure and/or cytokine storm resulting in death.
Line 53 ‘The virus can spread..’ should come earlier in the paragraph, i.e. just before Line 48 ‘The initial pathological changes…’
Line 78 modify use of Nucleocapsid protein (NP) abbreviation (rather than N protein elsewhere).

Experimental design

no comment

Validity of the findings

no comment

·

Basic reporting

*In spite of "SARS-CoV-2 virus" use only SARS-CoV-2
*Use uniform term for SARS-CoV-1 to SARS-CoV only throughout the manuscript

Experimental design

No comments

Validity of the findings

No comments

Additional comments

No comments

---

## Round 0.3 · accepted · Accept

Although I could not find your responses to the second reviewer's minor comments, I think the manuscript has been reasonably revised and is ready for acceptance. Congratulations!